# OpenReview forum: "Towards Building Model/Prompt-Transferable Attackers against Large Vision-Language Models"
_NeurIPS.cc/2025/Conference — NeurIPS 2025 spotlight_

### Official Review · Reviewer_7mhq · 2025-06-26

**Clarity:** 3
**Significance:** 4
**Originality:** 3
**Rating:** 5
**Confidence:** 5

**Summary:**

This paper investigates a critical issue of transferable adversarial attacks against LVLMs, by introducing a novel perspective of mutual information (MI). The core insight is that the informative dependence between the LVLM output and input images can be influenced by interactions between image and text prompts. With appropriate informative constraints between the disentangled adversarial/benign patterns of the image input and output text, the generated adversarial examples are proven to be more generalizable and harmful to unseen LVLMs and prompts. Extensive experiments are validated across various LVLM models and prompts with multiple LVLM datasets/tasks compared with existing LVLM attacks.

**Questions:**

1. The paper motivates that adversarial transferability is related to the MI values between adversarial/benign patterns and LVLM’s outputs. Although Figure 2 verifies this assumption, the authors should provide more discussion on it.
2. Target text selection: Using "I am sorry" as the target text may not truly represent a targeted attack. It would be valuable to include experiments with more specific target texts and report the success rates for such cases.
3. The authors are suggested to provide a case transfer-attack experiment across both models/prompts at the same time.
4. It seems that the proposed method achieves better transfer-attack performance but indicates reduced effectiveness in non-transfer attack scenarios. Although we know that it is due to the overfitting of previous attack methods, the authors could provide more discussion on this point.

**Ethical Concerns:**

["NO or VERY MINOR ethics concerns only"]

**Final Justification:**

I have carefully read the authors’ rebuttal and other reviewers’ comments. The responses have well-addressed my concerns. I believe the proposed MI adversarial constraints for improving the transferability demonstrates significant innovation to the LVLM security, while being theoretically grounded with information theory as mathematical support. I'm happy to see this work in the conference, and therefore I maintain my score: acceptance.

**Limitations:**

Yes

**Paper Formatting Concerns:**

None.

**Quality:**

4

**Strengths And Weaknesses:**

Strengths

1. The technical approach presented is novel and well-motivated. It proposes to improve the adversarial transferability across models/prompts using MI-based informative constraints. I think it is meaningful for the LVLM adversarial field.
2. Theoretical analyses are provided to support the proposed framework. The illustrated attack process is clear and easy to follow.
3. Sufficient experiments provided in both main paper and appendix validate the effectiveness of the proposed method.

Weaknesses

1. The paper motivates that adversarial transferability is related to the MI values between adversarial/benign patterns and LVLM’s outputs. Although Figure 2 verifies this assumption, the authors should provide more discussion on it.
2. Target text selection: Using "I am sorry" as the target text may not truly represent a targeted attack. It would be valuable to include experiments with more specific target texts and report the success rates for such cases.
3. The authors are suggested to provide a case transfer-attack experiment across both models/prompts at the same time.
4. It seems that the proposed method achieves better transfer-attack performance but indicates reduced effectiveness in non-transfer attack scenarios. Although we know that it is due to the overfitting of previous attack methods, the authors could provide more discussion on this point.

---

> ### Author Rebuttal · Authors · 2025-07-30
>
> **Q1: The paper motivates that adversarial transferability is related to the MI values between adversarial/benign patterns and LVLM’s outputs. Although Figure 2 verifies this assumption, the authors should provide more discussion on it.**
>
> **A1:** Thanks for your comment. Unlike previous works that implicitly optimize perturbations to destroy the image semantic representation, learning MI leads to an explicit optimization that emphasizes altering the comprehensive focus more on additive perturbation. The learned strong dependency between adversarial perturbation and attacker-chosen output allows the perturbation to learn and capture dominant features, thereby preserving adversarial harmfulness when transferring. This is the reason why we claim that a model/prompt-agnostic perturbation should always have more contributions/effects than the benign image to mislead the LVLMs during the reasoning process. Empirical findings on general transferable examples in Figure 2 of the paper also validate this motivation. More discussion can be found in Appendix E.
>
> **Q2: Target text selection: Using "I am sorry" as the target text may not truly represent a targeted attack. It would be valuable to include experiments with more specific target texts and report the success rates for such cases.**
>
> **A2:** Thanks for your suggestion. In fact, our attack is not limited to a fixed target text. We utilize the same "I am sorry" in the main experiments, as we want to keep a fair comparison for different ablation variants. We have provided ablation on different target texts in Table 5 of the paper and Table 10 of the appendix, which demonstrates that our attack is insensitive to diverse target texts. We further implement more target text experiments on LLaVA-1.5 (L), MiniGPT-4 (M), and BLIP-2 (B) in the table below, where our attack still achieves better transfer-attack performance under various target texts.
>
> |Target Text|LVLM Attack|(L to L) SS|(L to L) EM|(L to L) CC|(L to M) SS|(L to M) EM|(L to M) CC|(L to B) SS|(L to B) EM|(L to B) CC|
> |:----:|:----:|:----:|:----:|:----:|:----:|:----:|:----:|:----:|:----:|:----:|
> |"A man holding a big doughnut at a festival"|PGD|*0.916*|*91.2*|*91.2*|0.054|0.0|0.0|0.087|0.5|0.5|
> |"A man holding a big doughnut at a festival"|CroPA|*0.782*|*76.9*|*76.9*|0.040|0.0|0.0|0.087|0.0|0.0|
> |"A man holding a big doughnut at a festival"|UniAtt|*0.821*|*80.1*|*85.5*|0.204|17.8|21.7|0.291|18.9|28.3|
> |"A man holding a big doughnut at a festival"|**Ours**|*0.796*|*79.9*|*80.2*|**0.662**|**60.5**|**64.8**|**0.714**|**62.4**|**68.6**|
> |"A photo of a teddy bear on a skateboard in Times Square"|PGD|*0.903*|*89.8*|*91.2*|0.048|0.0|0.0|0.096|0.9|1.1|
> |"A photo of a teddy bear on a skateboard in Times Square"|CroPA|*0.786*|*78.2*|*78.2*|0.056|0.0|0.0|0.099|0.0|0.4|
> |"A photo of a teddy bear on a skateboard in Times Square"|UniAtt|*0.834*|*82.7*|*85.9*|0.216|18.6|21.5|0.285|19.6|25.4|
> |"A photo of a teddy bear on a skateboard in Times Square"|**Ours**|*0.779*|*79.8*|*79.8*|**0.671**|**61.2**|**66.9**|**0.695**|**62.7**|**67.7**|
> |"A beautiful bird with a black and white color in snow"|PGD|*0.907*|*90.1*|*90.1*|0.065|0.0|0.0|0.113|0.0|0.4|
> |"A beautiful bird with a black and white color in snow"|CroPA|*0.814*|*80.6*|*80.6*|0.052|0.0|0.0|0.091|0.4|0.8|
> |"A beautiful bird with a black and white color in snow"|UniAtt|*0.843*|*81.6*|*86.3*|0.229|18.3|23.1|0.287|20.7|29.6|
> |"A beautiful bird with a black and white color in snow"|**Ours**|*0.815*|*80.9*|*82.2*|**0.676**|**61.5**|**68.2**|**0.729**|**60.9**|**68.4**|
> |"Bunk bed with a narrow shelf sitting underneath it."|PGD|*0.876*|*88.3*|*88.3*|0.067|0.0|0.0|0.094|0.3|0.3|
> |"Bunk bed with a narrow shelf sitting underneath it."|CroPA|*0.785*|*77.2*|*79.6*|0.053|0.0|0.0|0.104|0.2|0.5|
> |"Bunk bed with a narrow shelf sitting underneath it."|UniAtt|*0.806*|*78.9*|*82.6*|0.214|16.7|22.3|0.274|20.1|26.8|
> |"Bunk bed with a narrow shelf sitting underneath it."|**Ours**|*0.799*|*79.5*|*81.2*|**0.621**|**57.4**|**63.5**|**0.645**|**58.4**|**65.0**|
> |"The people are gathered at the table for dinner."|PGD|*0.929*|*92.4*|*92.4*|0.056|0.0|0.0|0.105|0.5|0.8|
> |"The people are gathered at the table for dinner."|CroPA|*0.789*|*78.2*|*79.5*|0.046|0.0|0.0|0.088|0.0|0.3|
> |"The people are gathered at the table for dinner."|UniAtt|*0.831*|*82.0*|*85.9*|0.219|18.4|20.8|0.293|22.4|30.5|
> |"The people are gathered at the table for dinner."|**Ours**|*0.816*|*81.3*|*81.3*|**0.631**|**56.9**|**62.1**|**0.658**|**59.4**|**66.1**|
>
>
> **Q3: The authors are suggested to provide a case transfer-attack experiment across both models/prompts at the same time.**
>
> **A3:** Thanks for your suggestion. We evaluate the joint transferability across diverse models and prompts in the table below, which also demonstrates the effectiveness of our attack. We will add the experiments in the revision.
>
> |Transfer Attack|CroPA|UniAtt|**Ours**|
> |:----:|:----:|:----:|:----:|
> |LLaVA-1.5 to MiniGPT-4 (num=20)|0.042|0.115|**0.627**|
> |LLaVA-1.5 to MiniGPT-4 (num=60)|0.039|0.101|**0.608**|
> |MiniGPT-4 to LLaVA-1.5 (num=20)|0.050|0.189|**0.596**|
> |MiniGPT-4 to LLaVA-1.5 (num=60)|0.047|0.168|**0.594**|
>
>
> **Q4: It seems that the proposed method achieves better transfer-attack performance but indicates reduced effectiveness in non-transfer attack scenarios. Although we know that it is due to the overfitting of previous attack methods, the authors could provide more discussion on this point.**
>
> **A4:** Thanks for your comment. Our MI objective has some conflicts with the original attack objective during the adversarial learning process. As shown in Eq.(10) of the paper, we have two optimization goals: one is to optimize the perturbed image to achieve the attacker-chosen adversarial condition, and the other is to apply the informative constraints on disentangled adversarial/benign MI values of the dependency between the perturbed image and LVLM's output. Therefore, both items need to optimize the same perturbation noise of the perturbed image. To balance the adversarial effectiveness and transferability, our attack just slightly sacrifices some effectiveness but achieves significant improvement in transferability, which is more practical and scalable than the other attack methods. We will provide more discussion on this point in the revision.

---

> > ### Comment · Reviewer_7mhq · 2025-08-02
> >
> > I have carefully read the authors’ rebuttal and other reviewers’ comments. The responses have well-addressed my concerns of more discussion and experiments. I believe the proposed MI adversarial constraints for improving the transferability demonstrates significant innovation to the LVLM security, while being theoretically grounded with information theory as mathematical support. I'm happy to see this work in the conference, and therefore I maintain my score: acceptance.

---

> > > ### Author Response · Authors · 2025-08-02
> > >
> > > Thank you for your valuable response. We are very grateful for your high approval of our work. We will incorporate the rebuttal in detail into the revised version.

---

### Official Review · Reviewer_tw2d · 2025-06-26

**Clarity:** 2
**Significance:** 2
**Originality:** 3
**Rating:** 4
**Confidence:** 4

**Summary:**

This paper investigates a model- and prompt-transferable LVLM attack from an information-theoretic perspective. It then proposes an MI constraint, estimated by two networks in practice, to improve the transferability of adversarial examples by increasing the dependence of model output on the perturbation and decreasing the dependence of model output on the initial image.

**Questions:**

1. The overall optimization framework does not explicitly include the prompts. How does transferability improve when using the proposed MI constraints?
> In lines 221 to 228, the main idea is that the model's output depends more on the perturbation. However, a sharp turn without further explanation (like the phrase 'in this manner') suggests that the method becomes transferable to different models or prompts.

2. What is the relationship between the proposed method and other approaches that aim to improve the transferability of adversarial examples, such as augmentation-based and ensemble-based methods? This connection is not clearly explained in the related work. If the proposed method operates alongside existing methods, combining the two strategies in experiments could further enhance the effectiveness of this method as a 'plug and play' module. If the proposed method improves upon these approaches, then incorporating them into the baseline is necessary.

If the weakness can be fixed and the questions answered, I would like to improve my score. Right now, there are several flaws that lessen the effectiveness of this work below an acceptable level.

**Ethical Concerns:**

["NO or VERY MINOR ethics concerns only"]

**Final Justification:**

- The flaw in the theoretical analysis is solved by proposing reasonable assumptions and conditions in the rebuttal
- The motivation and how 'prompt transfer' is  achieved is solved by A3
- Still, the experiment is insufficient. The target text should be randomly chosen in the COCO caption set instead of being manually set to test the better transferability of the method

**Limitations:**

None.

**Paper Formatting Concerns:**

No.

**Quality:**

3

**Strengths And Weaknesses:**

**Strengths**

1. Clear motivation. This work aims to investigate and improve transferability through different models and prompts.
2. Adequate empirical studies. Ablations under different settings validate the generalization of this method.
3. The perspective from information theory is interesting.

**Weaknesses**

1. Typo. In Equ. (2), the left and right brackets do not match. The formula is hard to read, though the correct version can be found in the appendix.
2. The subsequent approximation step (lines 181-188) is problematic. Sepcifically, given $x_v^{adv} = x_v + \Delta$, if viewing $x_v^{adv}, x_v, \Delta$ as three random variables as in line 177, then $p(x_v^{adv})=p(x_v, \Delta)$ ***generally does not hold***, thus the two term $H(L'|x_v, \Delta), H(L', x_v^{adv})$ **cannot cancel out**. Even in the simplest case, where $x_v, \Delta$ is independent (which is  inaccurate, since the $\Delta$ depends on the initial image $x_v$ and the target), we have the PDF of $x_v^{adv}$ as  $f_{x_v^{adv}}(x)=\int_{-\infty}^{\infty} f_{x_v}(y) \cdot f_{\Delta}(x-y) dy$. In the dependent case, this is more complicated. Without further assumption, replacing $p(x_v^{adv})$ with $p(x_v, \Delta)$ is incorrect. Therefore, more than two MI terms are left after the approximation step.
3. The ablation study of different target texts is crucial yet insufficient. The metric used in Table 10 is missing, and only a few target texts are provided without comparison to other baselines. Without knowing the baseline performance, the advantage and overall effectiveness of the proposed method under this setting lack an anchor point.

---

> ### Author Rebuttal · Authors · 2025-07-30
>
> **Q1: Typo. In Equ. (2).**
>
> **A1:** Sorry for the confusion. We will correct the typos in Eq.(2) in the revision. The full Theorem and the proof can be found in Appendix B.
>
> **Q2: The subsequent approximation step (lines 181-188) is problematic.**
>
> **A2:** Thanks for your concern. First of all, we want to clarify that $x_v^{adv}, x_v, \Delta$ can **NOT** be viewed as three random and independent variables, as $\Delta$ is specifically optimized based on the input $x_v$ (the target is pre-defined and fixed). Therefore, the relation between $p(x_v^{adv})$ and $p(x_v, \Delta)$ should be dependently defined.
>
> Secondly, we have provided our assumption as the prior of this specific case in lines182-183 "each adversarial image can be decoupled into a unique corresponding relationship". That is, we assume that $\Delta, x_v^{adv}$ are bijections of $x_v$, i.e. $\Delta, x_v^{adv}$ are dependently and uniquely determined by $x_v$ and the decompositon of $x_v^{adv}$ is also unique. This assumption is based on: each $x_v$ generates a unique perturbation $\Delta$ determined by the attack algorithm like PGD, where prompt-agnostic perturbation $\Delta$ is computed by visual-solely contexts $\Delta = \epsilon \cdot \text{sign}\nabla_{x_v} L(f(x_v), y_{tar})$. Therefore, $\Delta$ can be taken as a function of $x_v$, denoted as $\Delta = g(x_v)$ that is uniquely determined by $x_v$. This leads to a unique adversarial image $x_v^{adv} = x_v + \Delta = x_v + g(x_v) = h(x_v)$, where $x_v^{adv}$ can also be taken as a function of $x_v$. Conversely, although $x_v^{adv}$ may exist other decompositions in a purely mathematical sense, in the adversarial generation context, the uniqueness of this decomposition can be guaranteed by the above attack protocol.
>
> Thirdly, before explaining why $p(x_v^{adv}) = p(x_v, \Delta) $ holds, as in Eq.(3) of the paper, we further claim that each $x_v,\Delta,x_v^{adv}$ is selected from a finite, countable set, making the selection process discrete. Therefore, we should utilize the probability mass function (PMF) instead of PDF in our specific case. Based on the above assumption, for discrete (Not continuous‌) variables $x_v,\Delta,x_v^{adv}$, the PMF of $x_v$ can denote as $p_{x_v}(x)=P(x_v=x)$. Since $\Delta$ is a bijection of $x_v$, there exists $y=g(x)$ such that $p_{x_v, \Delta}(x, y)=P(x_v=x, \Delta=y)=P(x_v=x)P(\Delta=y|x_v=x)=P(x_v=x)$. Similarly, since $x_v^{adv}$ is a bijection of $x_v$, there exists $z=h(x)$ such that the PMF of $x_v^{adv}$ can be derived as $p_{x_v^{adv}}(z)=P(x_v^{adv}=z)=P(h(x)=z)=P(x_v=x)$. Thus, we have $p_{x_v^{adv}}(z) = p_{x_v, \Delta}(x, y)$, and for any $y'\neq y, p_{x_v, \Delta}(x, y')=0$, meaning that $p(x_v^{adv}) = p(x_v, \Delta)$.
> Based on this, since $\Delta, x_v^{adv}$ are bijections of $x_v$, we have: $p(L'|x_v, \Delta)= p(L'|x_v, g(x_v))=p(L'|x_v)$ and $p(L'|x_v^{adv})=p(L'|h(x_v))=p(L'|x_v)$.
> Now, expanding $H(L'|x_v,\Delta)$:
> $$H(L'|x_v,\Delta)
> = - \sum_{L', x_v, \Delta} p(L', x_v, \Delta) \log  p(L'|x_v, \Delta)
> = - \sum_{x_v, \Delta} p(x_v, \Delta) \sum_{L'} p(L'|x_v, \Delta) \log  p(L'|x_v, \Delta) $$
> $$= - \sum_{x_v, \Delta = {g(x_v)}} p(x_v, \Delta) \sum_{L'} p(L'|x_v, \Delta) \log  p(L'|x_v, \Delta) + \underbrace{[- \sum_{x_v, \Delta  {\neq} {g(x_v)}} p(x_v, \Delta) \sum_{L'} p(L'|x_v, \Delta) \log  p(L'|x_v, \Delta)]} _ {=0}
> = -\sum_{x_v} p(x_v) \sum_{L'} p(L'|x_v) \log  p(L'|x_v).$$
> Similarly, $H(L'| x_v^{adv}) = -\sum_{x_v} p(x_v) \sum_{L'} p(L'|x_v) \log  p(L'|x_v).$
> Therefore, $H(L'| x_v^{adv})$ and $H(L'|x_v,\Delta)$ can be canceled out.
>
> **Q3: The ablation study of different target texts is crucial yet insufficient.**
>
> **A3:** The metric used in Table 10 is Semantic Similarity. We re-organize Table 10 with baseline comparison and more target texts across LLaVA-1.5 (L), MiniGPT-4 (M), and BLIP-2 (B) in the table below, with metrics of SemanticSimilarity (SS), ExactMatch (EM), and ConditionalContain (CC). It indicates that our attack achieves better transferability than baselines under different target texts. We will add these experiments in the revision.
>
> |Target Text|LVLM Attack|(L to L) SS|(L to L) EM|(L to L) CC|(L to M) SS|(L to M) EM|(L to M) CC|(L to B) SS|(L to B) EM|(L to B) CC|
> |:----:|:----:|:----:|:----:|:----:|:----:|:----:|:----:|:----:|:----:|:----:|
> |"A man holding a big doughnut at a festival"|PGD|*0.916*|*91.2*|*91.2*|0.054|0.0|0.0|0.087|0.5|0.5|
> |"A man holding a big doughnut at a festival"|CroPA|*0.782*|*76.9*|*76.9*|0.040|0.0|0.0|0.087|0.0|0.0|
> |"A man holding a big doughnut at a festival"|UniAtt|*0.821*|*80.1*|*85.5*|0.204|17.8|21.7|0.291|18.9|28.3|
> |"A man holding a big doughnut at a festival"|**Ours**|*0.796*|*79.9*|*80.2*|**0.662**|**60.5**|**64.8**|**0.714**|**62.4**|**68.6**|
> |"A photo of a teddy bear on a skateboard in Times Square"|PGD|*0.903*|*89.8*|*91.2*|0.048|0.0|0.0|0.096|0.9|1.1|
> |"A photo of a teddy bear on a skateboard in Times Square"|CroPA|*0.786*|*78.2*|*78.2*|0.056|0.0|0.0|0.099|0.0|0.4|
> |"A photo of a teddy bear on a skateboard in Times Square"|UniAtt|*0.834*|*82.7*|*85.9*|0.216|18.6|21.5|0.285|19.6|25.4|
> |"A photo of a teddy bear on a skateboard in Times Square"|**Ours**|*0.779*|*79.8*|*79.8*|**0.671**|**61.2**|**66.9**|**0.695**|**62.7**|**67.7**|
> |"A beautiful bird with a black and white color in snow"|PGD|*0.907*|*90.1*|*90.1*|0.065|0.0|0.0|0.113|0.0|0.4|
> |"A beautiful bird with a black and white color in snow"|CroPA|*0.814*|*80.6*|*80.6*|0.052|0.0|0.0|0.091|0.4|0.8|
> |"A beautiful bird with a black and white color in snow"|UniAtt|*0.843*|*81.6*|*86.3*|0.229|18.3|23.1|0.287|20.7|29.6|
> |"A beautiful bird with a black and white color in snow"|**Ours**|*0.815*|*80.9*|*82.2*|**0.676**|**61.5**|**68.2**|**0.729**|**60.9**|**68.4**|
> |"Bunk bed with a narrow shelf sitting underneath it."|PGD|*0.876*|*88.3*|*88.3*|0.067|0.0|0.0|0.094|0.3|0.3|
> |"Bunk bed with a narrow shelf sitting underneath it."|CroPA|*0.785*|*77.2*|*79.6*|0.053|0.0|0.0|0.104|0.2|0.5|
> |"Bunk bed with a narrow shelf sitting underneath it."|UniAtt|*0.806*|*78.9*|*82.6*|0.214|16.7|22.3|0.274|20.1|26.8|
> |"Bunk bed with a narrow shelf sitting underneath it."|**Ours**|*0.799*|*79.5*|*81.2*|**0.621**|**57.4**|**63.5**|**0.645**|**58.4**|**65.0**|
> |"The people are gathered at the table for dinner."|PGD|*0.929*|*92.4*|*92.4*|0.056|0.0|0.0|0.105|0.5|0.8|
> |"The people are gathered at the table for dinner."|CroPA|*0.789*|*78.2*|*79.5*|0.046|0.0|0.0|0.088|0.0|0.3|
> |"The people are gathered at the table for dinner."|UniAtt|*0.831*|*82.0*|*85.9*|0.219|18.4|20.8|0.293|22.4|30.5|
> |"The people are gathered at the table for dinner."|**Ours**|*0.816*|*81.3*|*81.3*|**0.631**|**56.9**|**62.1**|**0.658**|**59.4**|**66.1**|
>
> **Q4: The overall optimization framework does not explicitly include the prompts. How does transferability improve when using the proposed MI constraints?**
>
> **A4:** This is a good question. We have provided detailed discussions of "Transferability across prompts" and "Transferability across LVLMs" in Appendix E. We have also provided our motivation and empirical studies in lines 130-159 and Figure 2 of the paper. For more details, (1) As for transferability across prompts, *a prompt-agnostic perturbation should mislead the LVLM to focus less on the prompt inputs*. To achieve this, we explicitly enlarge the MI value solely between the LVLMs' output and the visual perturbations, to adjust the LVLMs' focus more on the visual perturbations and to weaken the LVLMs' reasoning dependence on the textual semantics, implicitly leading the LVLMs to ignore the effect of the prompt input. Compared to explicit constraints on prompts, this implicit design is more flexible and lightweight to achieve an effective cross-prompt attack. (2) As for transferability across LVLMs, *adversarial pattern should provide more semantic guidance than the benign one to fool LVLMs' visual encoder*. Therefore, we also introduce the MI constraints on the benign pattern, to help the whole perturbed image jump out of the traditional mixed adversarial-benign distribution. When feeding our adversarial examples into an unseen LVLM, the adversarial pattern will contribute more than the benign pattern in the distribution to mislead the LVLM's visual understanding.
>
> **Q5: What is the relationship between the proposed method and other approaches ... such as augmentation-based and ensemble-based methods?**
>
> **A5:** First, our proposed method operates alongside existing methods: Our method adversarially constrains the informative dependence between the benign/adversarial patterns of the input and the LVLM’s output to force the reasoning process to focus more on perturbations to enhance harmfulness during adversarial learning. However, augmentation-based methods improve transferability by applying input transformations during each iteration. Ensemble-based methods jointly attack multiple white-box models to generate generalizable perturbations. In summary, our method optimizes perturbations via pattern dependency, while existing augmentation- and ensemble-based methods focus on inputs' and models' diversity. *These approaches operate on different levels that are independent.*
>
> Second, according to your suggestion, we take our method as a 'plug and play' module to combine with existing methods: (1) we combine ours with augmentation method by transforming the inputs. (2) We combine ours with ensemble-based method by optimizing with multiple LVLMs. As shown in the table below, the results demonstrate that the combined strategies can further improve transferability, demonstrating the scalability of our proposed attack.
>
> |Attack|(L to L) SS|(L to L) EM|(L to L) CC|(L to M) SS|(L to M) EM|(L to M) CC|(L to B) SS|(L to B) EM|(L to B) CC|
> |:----:|:----:|:----:|:----:|:----:|:----:|:----:|:----:|:----:|:----:|
> |Ours|0.813|80.4|80.4|0.661|61.4|66.7|0.693|63.5|66.9|
> |Ours+Augmentation|0.818|81.2|81.9|0.674|63.5|67.6|0.705|65.3|67.7|
> |Ours+Ensemble|0.826|82.0|82.9|0.697|64.3|69.9|0.718|67.4|67.4|

---

> > ### Comment · Reviewer_tw2d · 2025-08-01
> >
> > Thanks for the author's clarification. The response to weaknesses and subsequent analysis help improve the paper's quality further. Thus, I have improved my score to 'borderline accept'.
> >
> > The rebuttal of the theoretical analysis part clarifies the issue. Still, it is not immediately obvious and may be misleading in the original submission (as seen in line 522, where A, B, C, and D denote four random variables without the specific conditions provided in the rebuttal). The assumption and conditions **are not explicitly exposed** but hidden inside lines. **It is not the reader's responsibility to find them if they are hidden**. The authors are encouraged to write formal assumptions and lemmas (like: Assumption 1, let A denote the finite set of original images, B denote the set of target images, ...) to clarify the specific conditions in the revised manuscript, to make the analysis complete and rigorous.

---

> ### Author Response · Authors · 2025-08-01
>
> We sincerely thank you for the helpful comment and the score improvement.
>
> In the revised version, we will incorporate the detailed explanations from the rebuttal and present them as formal assumptions and lemmas to enhance the clarity and rigor of the theoretical analysis.

---

### Official Review · Reviewer_ZrB4 · 2025-06-27

**Clarity:** 3
**Significance:** 3
**Originality:** 4
**Rating:** 5
**Confidence:** 4

**Summary:**

The paper introduces a novel attack method for Large Vision-Language Models (LVLMs) that enhances the transferability of adversarial examples across different models and prompts. Considering enlarging/decreasing the mutual information between outputs and the disentangled adversarial/benign patterns of input images helps to generate more agnostic perturbations for misleading LVLMs’ perception with better transferability, the paper proposes a novel perspective of information theory to improve the LVLM attacks’ transferability. Experiments across four LVLMs (LLaVA-1.5, MiniGPT-4, BLIP-2, InstructBLIP) and three datasets (DALL-E, SVIT, VQAv2) demonstrate significant improvements in cross-model and cross-prompt attack success rates and semantic similarity metrics.

**Questions:**

The reviewer's main concerns are these points and questions outlined in the weaknesses section. The reviewer suggests that the authors focus on these points in the rebuttal.

**Ethical Concerns:**

["NO or VERY MINOR ethics concerns only"]

**Final Justification:**

The authors have addressed all my concerns with quantitative evidence and clear explanations.

**Limitations:**

Limitations and potential negative impacts have been sufficiently discussed in the paper.

**Paper Formatting Concerns:**

There is no formatting issue.

**Quality:**

4

**Strengths And Weaknesses:**

**Strengths**

This paper addresses a significant gap in the field of adversarial attacks on LVLMs, offering a more practical and versatile approach compared to existing model/prompt-specific methods. The transferable nature of the attack and its ability make it particularly relevant for real-world applications and raise important questions about the security of LVLMs.

- Using Information theory to address the transferability of LVLM is novel.
- The proposed method is technically sound and comprehensive experiments demonstrate the empirical performance.
- The safety concerns of LVLMs exposed in this paper could potentially have a large impact on the AI safety field.

**Weaknesses**

The reviewer finds that this paper is novel and provides valuable insights to the community. However, there are several areas where the paper could be improved:

- There are various existing LVLM attacks are re-implemented for comparison. Do they re-implemented in the same setting for fair comparisons? More details are required.
- How to train the two MI estimation networks? Will they introduce additional noticeable costs? The authors should provide more corresponding analysis.
- A commonly implemented image-level universal setting should also be experimented.
- Two widely used baselines should be implemented for comparisons: “On evaluating adversarial robustness of large vision-language models” and “How Robust is Google's Bard to Adversarial Image Attacks?”
- The authors did a good job by providing sufficient discussion and experiments in the appendix. The reviewer suggests including the justification of fair transfer attacks in the main paper.

---

> ### Author Rebuttal · Authors · 2025-07-30
>
> **Q1: There are various existing LVLM attacks are re-implemented for comparison. Do they re-implemented in the same setting for fair comparisons? More details are required.**
>
> **A1:** Thanks for your concern. In fact, all LVLM attack baselines are re-implemented in the same setting for experimental comparison, including: (1) Consistent Model Selection: All methods are implemented and evaluated on the same LVLM models, such as LLaVA-1.5, MiniGPT-4, BLIP-2, and InstructBLIP. (2) Consistent Tasks and Datasets: All methods are evaluated on the same three types of tasks (image captioning, image classification, and VQA), the same datasets (DALL-E, SVIT, and VQAv2), and the same target texts. That is, all methods optimize adversarial examples based on the same image-prompt pairs and target texts. (3) Consistent Attack Constraints: All methods utilize the same perturbation bound $\epsilon=16/255$, the same epoch number, and the same step size $\alpha=16/epoch$. (4) Consistent Evaluation Metrics: All methods are evaluated using the same three metrics—semantic similarity (SS), Exact Match (EM), and Conditional Contain (CC). And the results of each metric are averaged across the three tasks for comparison. Therefore, our baseline comparison is fair. We will provide more details in the revision.
>
>
> **Q2: How to train the two MI estimation networks? Will they introduce additional noticeable costs? The authors should provide more corresponding analysis.**
>
> **A2:** The two MI estimation networks are initialized with the same architecture and trained separately. Specifically, $C_{\psi}$ is implemented as a light two-layer convolutional neural network, while $T_{\omega}$ simply incorporates an attention mechanism, $1\times 1$ convolutional blocks, and residual connections. For training, we first utilize the selected adversarial examples generated by the PGD attack. We then feed the same prompt with benign image ${x}_v$ and adversarial image ${x}^{adv}_v$ into the LVLM to obtain the corresponding logits outputs ${L}$ and ${L}^{'}$. The tuple $({x}_v, {L}, {L}^{'})$ is used as input to train the benign MI estimation network based on Equation (7) in the paper, while the tuple $({x}^{adv}_v - {x}_v, {L}, {L}^{'})$ is used to train the adversarial MI estimation network following Equation (6). We train both networks using the Adam optimizer for 100 epochs, with an initial learning rate of 0.01 that decays by a factor of 0.5 every 10 epochs. These MI estimation networks are very lightweight and just cost about 26mins. Therefore, it will not introduce much time cost to the average time speed of single adversarial sample generation. We will provide corresponding analysis in the revision.
>
> **Q3: A commonly implemented image-level universal setting should also be experimented.**
>
> **A3:** Thanks for your comment. Our work can be easily extended to an image-level universal setting by optimizing the perturbation across images, as shown in the table below, where our attack still achieves competitive performance.
>
> |Variant|GPT-4o|Claude-3.5|
> |:----:|:----:|:----:|
> |(universal) From MiniGPT-4 to|0.579|0.605|
> |(non universal) From MiniGPT-4 to|0.608|0.620|
>
>
> **Q4: Two widely used baselines should be implemented for comparisons: “On evaluating adversarial robustness of large vision-language models” and “How Robust is Google's Bard to Adversarial Image Attacks?”**
>
> **A4:** Thanks for your suggestions. As for more baseline comparisons, we re-implement your mentioned two papers, “On evaluating adversarial robustness of large vision-language models” (Zhao et al.) and “How Robust is Google's Bard to Adversarial Image Attacks?” (Dong et al.) into targeted attack transferred from MiniGPT-4 to GPT-4o and Claude-3.5 in the table below, where our attack still achieves better transferability.
>
> |Attack Method|GPT-4o (SS)|GPT-4o (EM)|GPT-4o (CC)|Claude-3.5 (SS)|Claude-3.5 (EM)|Claude-3.5 (CC)|
> |:----:|:----:|:----:|:----:|:----:|:----:|:----:|
> |Zhao et al.|0.161|12.8|17.0|0.189|13.6|17.9|
> |Dong et al.|0.129|10.2|12.4|0.154|11.3|15.7|
> |**Ours**|**0.608**|**44.7**|**51.3**|**0.620**|**48.6**|**56.5**|
>
>
> **Q5: The authors did a good job by providing sufficient discussion and experiments in the appendix. The reviewer suggests including the justification of fair transfer attacks in the main paper.**
>
> **A5:** Thanks for your suggestion. We will move the transfer-attack justification (F.4 Justification of Our Transfer Attack) from the appendix to the main paper in the revision.

---

> > ### Comment · Area_Chair_1pXy · 2025-08-05
> >
> > Dear Reviewer ZrB4,
> >
> > When you have a moment, please take a look at the authors’ rebuttal and finalize your score.
> >
> > Best wishes, AC

---

> > ### Comment · Reviewer_ZrB4 · 2025-08-07
> >
> > The authors have addressed all my concerns with quantitative evidence and clear explanations. After seeing the comments of other reviewers and the rebuttals, I think this work brings novel insight of information theory to the LVLM’s transfer attack with MI decomposition, estimation, and optimization. The proposed method also achieves impressive performances. Based on these points, I recommend the acceptance of this paper.

---

> > > ### Author Response · Authors · 2025-08-07
> > >
> > > We would like to thank the reviewer for responding to our rebuttal. It is great to know that your concerns have been addressed. Thank you once again for your valuable contributions.

---

### Official Review · Reviewer_MLbA · 2025-07-01

**Clarity:** 3
**Significance:** 2
**Originality:** 2
**Rating:** 4
**Confidence:** 5

**Summary:**

This paper addresses the critical challenge of transferability in adversarial attacks against LVLMs, noting that existing methods often overfit to specific models and prompts, limiting their practical effectiveness. The authors propose a novel approach from an information theory perspective. The core idea is to disentangle the adversarial perturbation from the benign image content and explicitly manipulate the model's reasoning process by controlling the mutual information between the input components and the model's output.  Extensive experiments on multiple LVLMs and datasets, including transfers to black-box models, demonstrate that the proposed method significantly outperforms existing attacks in both cross-model and cross-prompt scenarios, underscoring the effectiveness of their approach.

**Questions:**

Given the impressive transfer results to Qwen2-VL, how does the proposed method's transferability perform when targeting another architecturally distinct and powerful model, such as InternVL?

**Ethical Concerns:**

["NO or VERY MINOR ethics concerns only"]

**Final Justification:**

The authors have addressed my concerns. If the additional experiments are incorporated into the main body, it would enhance the completeness and rigor of the paper, thereby meeting the standard for acceptance.

**Limitations:**

No.

**Quality:**

3

**Strengths And Weaknesses:**

Strengths

1.The proposed method achieves state-of-the-art performance in transfer attacks across multiple LVLM benchmarks.
2.The paper provides a comprehensive set of experiments, including extensive comparisons and ablation studies, which effectively demonstrate the method's strong performance.

Weaknesses

1.The analysis of robustness against defenses is incomplete. The experiments only demonstrate resilience in a white-box setting and lack an evaluation of how the proposed cross-model and cross-prompt attacks perform when transferred to defended target models.
2.The study lacks transferability experiments against other architecturally distinct model families, such as InternVL, which would provide a more complete assessment of the method's generalizability.

---

> ### Author Rebuttal · Authors · 2025-07-30
>
> **Q1: The analysis of robustness against defenses is incomplete. The experiments only demonstrate resilience in a white-box setting and lack an evaluation of how the proposed cross-model and cross-prompt attacks perform when transferred to defended target models.**
>
> **A1:** Thanks for your comment. We provide a more detailed analysis of robustness against defenses in the table below, where we show the cross-model/prompt attacks' performance under various defenses. In particular, we also introduce two new defense mechanisms for defended target evaluation: one improves the CLIP component in BLIP-2, LLaVA-1.5, MiniGPT-4 and InstructBLIP models with a defended FARE model (2024 ICML) of paper "Robust CLIP: Unsupervised Adversarial Fine-Tuning of Vision Embeddings for Robust Large Vision-Language Models", and the other uses a defended DPS model (2025 ICML) of paper "Defending LVLMs Against Vision Attacks through Partial-Perception Supervision" to embed the input of the BLIP-2, LLaVA-1.5, MiniGPT-4 and InstructBLIP models. The experimental results demonstrate that our cross-model/prompt attack still achieves better transferability compared to baselines under various defense methods, indicating our robustness. We will add these defense experiments in the revision.
>
> |Defense|Attack|BLIP-2 to BILP-2|BLIP-2 to LLaVA-1.5|BLIP-2 to MiniGPT-4|BLIP-2 to InstructBLIP|Across Prompts (Num=20)|
> |:----:|:----:|:----:|:----:|:----:|:----:|:----:|
> |Randomization|PGD|0.213|0.014|0.019|0.042|0.169|
> |Randomization|CroPA|0.281|0.046|0.041|0.062|0.188|
> |Randomization|UniAtt|0.489|0.156|0.121|0.096|0.336|
> |Randomization|**Ours**|**0.510**|**0.466**|**0.439**|**0.254**|**0.463**|
> |JPEG Compression|PGD|0.348|0.023|0.018|0.046|0.340|
> |JPEG Compression|CroPA|0.476|0.054|0.051|0.051|0.411|
> |JPEG Compression|UniAtt|0.653|0.251|0.213|0.135|0.474|
> |JPEG Compression|**Ours**|**0.676**|**0.613**|**0.560**|**0.429**|**0.618**|
> |Diffusion Restoration|PGD|0.258|0.021|0.020|0.043|0.217|
> |Diffusion Restoration|CroPA|0.349|0.044|0.046|0.048|0.275|
> |Diffusion Restoration|UniAtt|0.321|0.122|0.109|0.084|0.318|
> |Diffusion Restoration|**Ours**|**0.535**|**0.464**|**0.443**|**0.279**|**0.485**|
> |FARE (2024 ICML)|PGD|0.246|0.023|0.017|0.041|0.206|
> |FARE (2024 ICML)|CroPA|0.316|0.046|0.047|0.064|0.258|
> |FARE (2024 ICML)|UniAtt|0.408|0.136|0.113|0.104|0.321|
> |FARE (2024 ICML)|**Ours**|**0.547**|**0.468**|**0.449**|**0.286**|**0.489**|
> |DPS (2025 ICML)|PGD|0.319|0.025|0.024|0.048|0.274|
> |DPS (2025 ICML)|CroPA|0.382|0.051|0.046|0.069|0.293|
> |DPS (2025 ICML)|UniAtt|0.464|0.154|0.120|0.097|0.325|
> |DPS (2025 ICML)|**Ours**|**0.576**|**0.481**|**0.454**|**0.291**|**0.512**|
>
>
>
> **Q2: The study lacks transferability experiments against other architecturally distinct model families, such as InternVL, which would provide a more complete assessment of the method's generalizability.**
>
> **A2:** Thanks for your suggestion. We have provided the transfer-attack experiments for architecturally distinct models GPT-4o, Claude-3.5 in Table 3 of the paper, and InternVL models in Table 6 of the appendix. To provide more detailed transfer-attack experiments on architecturally distinct model families, we comprehensively evaluate the attacks' transferability among MiniGPT-4 (EVA-CLIP-ViT-g-14), Qwen2-VL (CLIP-ViT-bigG), InternVL (InternViT-300M-448px-V2_5), and Gemma-3 (SigLIP-ViT) in the table below. It shows that our attack is more generalizable to architecturally distinct LVLMs compared to other attacks, demonstrating our great transferability. We will provide corresponding experiments in the revision.
>
> |Source Model|LVLM Attack|MiniGPT-4|Qwen2-VL|InternVL|Gemma-3|
> |:----:|:----:|:----:|:----:|:----:|:----:|
> |MiniGPT-4|PGD|*0.823*|0.034|0.042|0.031|
> |MiniGPT-4|CroPA|*0.955*|0.042|0.051|0.045|
> |MiniGPT-4|UniAtt|*0.830*|0.128|0.145|0.139|
> |MiniGPT-4|**Ours**|*0.860*|**0.639**|**0.610**|**0.591**|
> |Qwen2-VL|PGD|0.054|*0.712*|0.069|0.051|
> |Qwen2-VL|CroPA|0.076|*0.763*|0.093|0.083|
> |Qwen2-VL|UniAtt|0.199|*0.820*|0.224|0.241|
> |Qwen2-VL|**Ours**|**0.622**|*0.792*|**0.691**|**0.612**|
> |InternVL|PGD|0.051|0.074|*0.811*|0.043|
> |InternVL|CroPA|0.085|0.108|*0.823*|0.079|
> |InternVL|UniAtt|0.212|0.207|*0.829*|0.224|
> |InternVL|**Ours**|**0.683**|**0.676**|*0.847*|**0.627**|
> |Gemma-3|PGD|0.063|0.059|0.067|*0.842*|
> |Gemma-3|CroPA|0.088|0.104|0.117|*0.815*|
> |Gemma-3|UniAtt|0.237|0.221|0.234|*0.833*|
> |Gemma-3|**Ours**|**0.641**|**0.618**|**0.620**|*0.819*|

---

> > ### Comment · Reviewer_MLbA · 2025-08-05
> >
> > Thank you to the authors for their feedback. My concerns have been resolved, and I hope the authors will keep their promise to include the experiments in the final version of the paper.

---

> ### Author Response · Authors · 2025-08-05
>
> Thank you for your valuable response. We guarantee that we will incorporate the rebuttal in detail into the revised version.

---

### Decision · Program_Chairs · 2025-09-17

**Decision:**

Accept (spotlight)

**Comment:**

This work explore the transferability of adversarial examples in MLLMs from mutual information perspective. Compared to previous work studying cross-prompt transferable attacks [8, 10], this work propose a novel way to further mitigate the overfitting of adversarial perturbation.

All reviewers have positive feedback after rebuttal, I recommend this paper for acceptance. The authors are encouraged to include the discussion into the revision.